# Tetracycline Degradation by Peroxydisulfate Activated by Waste Pulp/Paper Mill Sludge Biochars Derived at Different Pyrolysis Temperature

**Baowei Zhao * and Juanxiang Zhang**

School of Environmental and Municipal Engineering, Lanzhou Jiaotong University, Lanzhou 730070, China; 120210084@taru.edu.cn
* Correspondence: baoweizhao@mail.lzjtu.cn; Tel.: +86-931-49-55-760

**Abstract:** The technique of using biochar-based catalysts in persulfate activation is a promising alternative to remov emerging and refractory pollutants (e.g., tetracycline-) in wastewater. However, the situation of biochars derived from waste pulp/paper mill sludge is still unclear. The pulp/paper mill sludge biochars (SBC300, SBC500, and SBC700) were obtained and characterized at pyrolysis temperatures of 300, 500, and 700 °C. Tetracycline degradation using peroxydisulfate activated by SBCs was investigated. The results demonstrated the removal efficiencies of tetracycline in SBC300-, SBC500- and SBC700-peroxydisulfate systems, which increased with the pyrolysis temperatures and were 4.3, 4.8, and 5.0 times that of a system with peroxydisulfate alone. The experiments of free radical quenching, singlet oxygen quenching, and electrochemistry indicated that the degradation of tetracycline in SBC-peroxydisulfate systems was mainly not a free radical pathway, but a non-radical pathway. Singlet oxygen ($^1O_2$) and electron transfer could play main roles in the degradation removal of tetracycline. The removal efficiencies of tetracycline in the SBC-peroxydisulfate systems could be up to 96.0% (SBC700-peroxydisulfate) under the optimum dosage of SBC, the molar ratio of peroxydisulfate to tetracycline and the solution pH value. The results indicate that a SBC700-peroxydisulfate system could be an effective "trash-to-treasure" treatment technique for wastewater containing antibiotics.

**Keywords:** waste pulp/paper mill sludge; biochar; peroxydisulfate; tetracycline; degradation





## 1. Introduction

Tetracycline (TC) is a broad-spectrum antibiotic [1]. It is not only used in clinical medicine to control human diseases, but also as veterinary drug and growth promoter in the livestock and poultry breeding industry [2,3]. It is estimated that about 20–90% of antibiotics are excreted into the environment unaltered or as transformation products through animal feces and urine [4]. Due to its poor bioavailability, the presence of TC in the aqueous environment can result in the issues of antibiotic resistant bacteria (ARB) and antibiotic resistance genes (ARGs) [5,6], which significantly reduce the remedial potential of antibiotics against human and animal bacteria and viruses [7].

Recently, various techniques such as adsorption [8,9], membrane filtration [10], photocatalytic degradation [11], biodegradation [12], and advanced oxidation processes (AOPs) [13,14] have been investigated to remove TC in water. Among these means, AOPs based on sulfate radical ($SO_4^{\bullet-}$) are followed with more and more interest due to the following advantages: (i) compared with the redox potential of $HO^\bullet$ (1.9–2.7 V), $SO_4^{\bullet-}$ possesses a higher one (2.5–3.1 V) [15]; (ii) the half-life of the sulfate radical $SO_4^{\bullet-}$ (30–40 μs) is larger than that of the hydroxyl radical $HO^\bullet$ ($\leq 1$ μs) [16,17]; (iii) persulfate (PS) is also cheaper than other oxidants such as $H_2O_2$ ($0.74/kg of PS vs $1.5/kg of $H_2O_2$) [18]. Accordingly, different activation methods, such as ultrasound and ultraviolet light, Fe, Co, and Mn-based catalysts, and acidic, alkaline, and phenol chemicals, have been used to motivate PS to produce

active radicals [19–21]. Although they are effective processes, there are some limitations in energy consumption, high expenditure, and the danger of heavy metals being released [18].

In recent years, the low-cost, easily accessible, and environmentally friendly biochars, which have a large specific surface area, plentiful pore structure, and affluent oxygen-containing functional groups, have been proven to be potential catalysts for PS activation [22]. Municipal sewage sludge (SS) is a waste engendered by wastewater treatment. Recently, it has been shown that converting SS into biochar and using it as a feasible adsorbent and catalyst substance could supply an alternative for both SS disposal and contaminant removal [22–25]. SS-derived biochars exhibit various characteristics and properties, including radical and non-radical oxidation processes, resulting in the various mechanisms for PS activation to degrade organic refractory pollutants. It was found that $SO_4^{\bullet-}$ and $HO^\bullet$ played the main roles in the degradation removal of 4-chlorophenol [26]; $^1O_2$ played a critical role in the degradation of bisphenol A and sulfamethoxazole [25,27], sulfathiazole was degraded by a non-radical route electron transfer [28], and $SO_4^{\bullet-}$, $HO^\bullet$, and $^1O_2$ contributed to triclosan degradation [29]. Compared with SS, the waste pulp/paper mill sludge (PPS) is characterized by a high content of hemicellulose, cellulose, and lignin [30], which could be beneficial to the derived biochars in regard to improving the production of active oxygen particulates ($SO_4^{\bullet-}$, $HO^\bullet$, $O_2^{\bullet-}$, and $^1O_2$) and would have an important effect on the catalytic degradation removal of organic contaminants [31–33]. Moreover, it is shown that PPS-derived biochars are prominent adsorbents for many contaminants from wastewater, such as metals, pharmaceuticals, organics, and dyes [34–38]. Our previous study [39] showed that the adsorption of TC by PPS-derived biochars was effective, and the adsorption mechanisms are partition, electrostatic attraction, hydrogen bonding, π-π electro donor-acceptor, and ion exchange. However, to our knowledge, there have been few studies on PPS-derived biochars for PS activation to degrade organic contaminants (e.g., antibiotics) in water, and the degradation mechanisms are also unclear.

In this study, the biochars were prepared using PPS as biomass through pyrolysis at different temperatures (300, 500, and 700 °C) (SBC300, SBC500, and SBC700), the basic physical-chemical properties of the biochars were characterized, and the peroxydisulfate (PDS) activation performance by the biochars for TC degradation together with its mechanisms and optimum conditions was investigated. The objectives are (i) to identify the effects of the pyrolysis temperatures on the biochar's properties and removal efficiencies of TC; (ii) to probe into the main routes for the degradation of TC; and (iii) to illustrate the feasibility of SBC-based catalysts in PDS activation to remove TC in water. The results imply that an SBC-PDS system could be an effective, environmentally friendly, and low-cost strategy for the clean-up of antibiotics in wastewater.

## 2. Materials and Methods

### 2.1. Chemicals and Materials

Tetracycline (TC), methanol (MeOH), *tert*-butanol (TBA), sodium peroxydisulfate (PDS), and L-histidine were purchased from Tianjin Damao Chemical Reagent Factory (Tianjin, China). All the chemical reagents were analytically pure, and the solutions were prepared with deionized water. Waste pulp/paper mill sludge (PPS) was sampled from the Wastewater Treatment Plant of Paper Mill in Jingning, Gansu Province, China. The sludge was naturally dried, ground, and passed through a 60-mesh sieve. Then, the sludge powder was put into a muffle furnace (HWL-12XC, Shandong Huawei Luye Company, China) and pyrolyzed at 300, 500, and 700 °C for 6 h by oxygen-limiting and temperature-controlling methods. After cooling to ambient temperature, the biochars were taken out, stored, and recorded as SBC300, SBC500, and SBC700, respectively.

### 2.2. Characterization of Biochar

The values of point of zero charge (pH$_{pzc}$) of SBCs were measured by acid-base potential titration method. The surface area-pore analyzer (ASAP2020, Micromeritics, Norcross, GA, USA) was utilized to measure the specific surface areas by N$_2$ sorption.

The compositions of C, N, H and O elements in SBCs were measured by Automatic Elemental Analyzer (vario EL, Elementar, Langenselbold, Germany). X-ray diffraction (XRD) graphs were plotted using a diffraction meter (Rigaku D/max-2400, Japan) with Cu K$_\alpha$ radiation at 10.154056 nm and 2θ from 5 to 90°. Electrical conductivity was measured using electrical conductivity meter (Five Easy, METTLER TOLEDO, Columbus, OH, USA) with 1:20 of biochar: distilled water mixture [40]. The oxygen-containing functional groups of SBCs were identified by Fourier Transform Infrared Spectroscopy (FTIR, Nexus 870, ThermoFisher, Waltham, MA, USA).

### 2.3. Catalytic Degradation Experiments

The catalytic performances of SBCs were conducted in a thermostatic shaker (CHA-S Shaker, Jintan Danyang Instrumental Company, Changzhou, China) with a shaking rate of 150 rpm. A total of 50 mg of SBC was evenly dispersed into 50 mL of 20 mg/L TC solution. Then, a certain proportion of PDS was added into the system to make the molar ratio of PDS to TC 100:1. The mixture was shaken at 25 °C for certain time. Then, 5 mL of the mixture was taken out, combined with MeOH, and filtered through 0.45 μm filter-membrane to detect the concentration of TC. The concentration of TC was analyzed using the Ultraviolet and Visible Spectrophotometer (UV-SP, Shanghai Spectrum Instrument Co., Shanghai, China) at 356 nm [41]. An amount of 0.1 M NaOH or HCl solution was used to adjust the pH value of the mixture. When the effects of SBC dosage, molar ratio of PDS to TC, and solution pH value on degradation were concerned, respectively, one factor was accordingly changed while the others were kept the same. When the biochar reuse was concerned, the mixture after degradation test was filtered through 0.45 μm membrane. The filtered biochar was repeatedly rinsed with water and dried at 40 °C in an oven. Then, its catalytic performance was tested according to the steps mentioned above (run 1). Four runs were performed to test the catalytic performances of SBCs. All the experiments were conducted in triplicate, and the mean values were calculated for analysis.

### 2.4. Experiments for Mechanism Exploration

The steps for free radical quenching and singlet oxygen quenching experiments were designed similar to those in catalytic degradation experiments at large. For the free radical quenching experiments, the molar ratios of scavenger, methanol (MeOH), or *tert*-butyl alcohol (TBA) to TC were set as 0:1, 1000:1, 2000:1, and 5000:1, respectively. For the singlet oxygen quenching experiments, L-histidine was added to the concentrations of 0, 5, 10, and 15 mmol/L in the systems. The electrochemical measurements, linear-sweep voltammograms (LSV), and electrochemical impedance spectroscopy (EIS) were performed on an electrochemical workstation (CHI660E, Shanghai Zhenhua, China) with a three-electrode cell, i.e., biochar as working electrode, platinum wire electrode as auxiliary cell, and saturated calomel electrode as reference one.

## 3. Results and Discussion

### 3.1. Characterization of SBCs

Table 1 shows the main elemental composition, pH$_{pzc}$ values, and BET-specific surface areas of SBCs obtained at different pyrolysis temperatures. The specific areas of biochars were 7.12, 32.82, and 37.80 m$^2$/g, respectively. The specific surface area of SBC increased with the increased pyrolytic temperature. Because PPS possesses abundant cellulose and lignin [30], the internal pyrolysis of these organic compounds was intensified at high temperatures, which led to the breaking of intermolecular and intramolecular chemical bonds and the releasing of small molecular substances (e.g., CH$_4$ and CO$_2$). The escape of generated gas and volatile components caused the creation of pore structure in the solid. Upon raising the pyrolytic temperature, the tar was gradually decomposed, the pore formation rate was accelerated, and the specific surface area increased [30]. This could be the reason for the formation of a large specific area [39]. Generally, the ratios of H/C, O/C, and (N+O)/C are used to characterize the aromaticity, hydrophilicity, and

polarity of a biochar. The decreasing ratios of H/C indicate that the aromaticity of SBCs rose with pyrolitic temperatures, which could result in strong electron providing-accepting effects with aromatic pollutants through the $\pi$-electron conjugation system [18]. It has been indicated that some important functional groups consisting of carbon, hydrogen, and oxygen atoms (for example, –OH and C=O) are the active sites of general reactive free radicals, such as hydroxyl radicals and sulfate radicals. Also, it has been shown that nitrogen atoms (pyridinic N and graphitic N) could improve the electron transfer at the interface of carbon-based catalysts [42]. However, the hydrophilicity and polarity of SBC500 are the strongest. When the pyrolysis temperature increased, the values of $pH_{pzc}$ SBCs increased.

**Table 1.** Basic physical-chemical properties of biochars [39].

| Biochar | Elemental Composition (wt%) | | | | H/C | O/C | (N+O)/C | $pH_{pzc}$ | Specific Area (m$^2$/g) |
|---|---|---|---|---|---|---|---|---|---|
| | C | N | H | O | | | | | |
| SBC300 | 19.98 | 1.64 | 1.95 | 19.84 | 1.17 | 0.74 | 0.82 | 7.98 | 7.12 |
| SBC500 | 15.96 | 0.67 | 0.43 | 19.93 | 0.32 | 0.94 | 0.97 | 9.25 | 32.82 |
| SBC700 | 14.11 | 0.45 | 0.29 | 15.99 | 0.25 | 0.85 | 0.88 | 9.87 | 37.80 |

XRD was conducted to identify the main crystalline phases of SBCs. Figure 1a reveals the XRD patterns of the SBC samples. There is no fundamental change in the phase composition among the specific SBCs. The main diffraction peaks at 2$\theta$ = 29.4° of the SBCs represent the microcrystalline structure of $CaCO_3$ (calcite). This shows that there are more Ca components in the SBCs, which mainly exist in the form of calcium carbonate. Also, it can be seen that the contents of calcium carbonate in the SBCs decreased when the pyrolytic temperature increased. Some weak diffraction peaks occur at 2$\theta$ = 39.5° and 47.9°, which might be the microcrystalline occurrence of $CaSO_4$. Furthermore, it is found that SBCs have relatively weak diffraction peaks at 2$\theta$ = 36° and 43°, which might be the quartz phase, such as $SiO_2$ [43]. The diffraction peaks at 43.5° and 23.3° are representative of crystalline carbon and amorphous carbon, respectively [44].

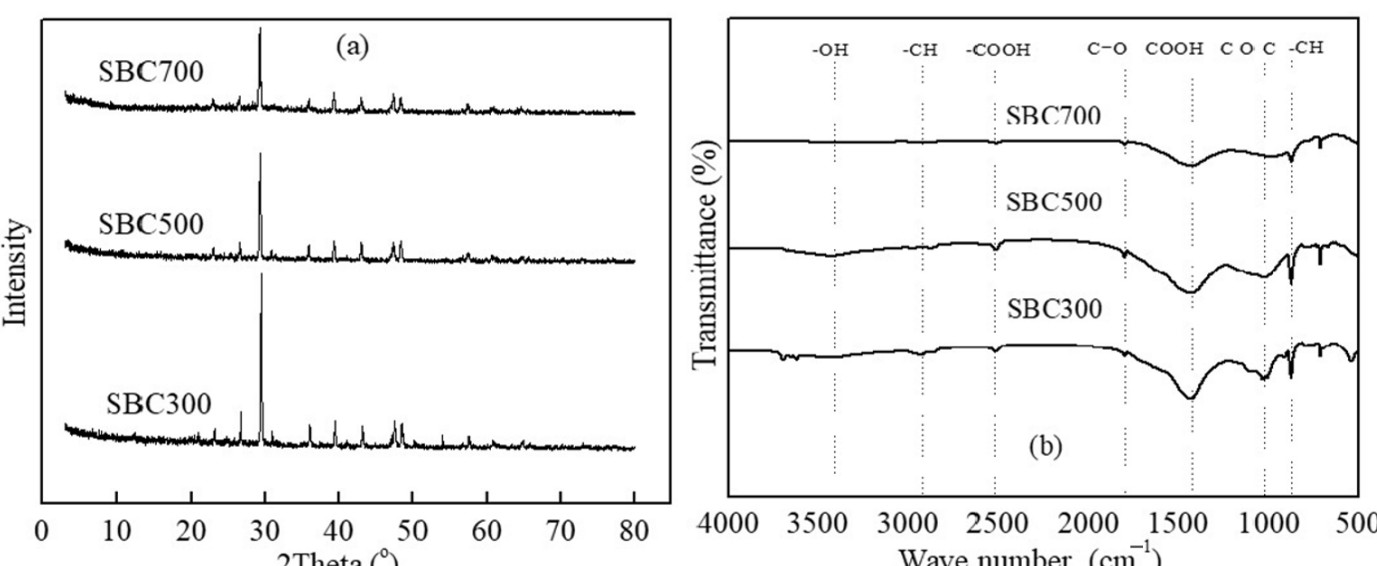

**Figure 1.** XRD patterns (**a**) and FTIR spectra (**b**) of SBCs.

As shown in Figure 1b, with the increased pyrolytic temperature, the type and amount of oxygen-containing function groups of SBCs gradually decreased, especially for the –COOH and –C–O–C ranging around 1431 cm$^{-1}$ and 1007 cm$^{-1}$. A similar change pattern is found in the peak around 1790 cm$^{-1}$ and 870 cm$^{-1}$ which might belong to the vibration

of –C=O and the –CH in SBC500 and SBC700. This phenomenon might be attributed to the ongoing carbonization of PPS under higher pyrolytic temperatures. Due to the dehydration of lignin and celluloses, the peak of O–H stretching decreased rapidly after 500 °C [45]. In addition, the weakened absorption peaks of aliphatic –CH also indicated the formation of aromatic structure.

### 3.2. Degradation Performance

In order to assess the catalytic ability of SBCs for PDS activation to degrade TC, the degradation processes had been conducted including the single PDS and combined SBC-PDS systems. Figure 2 shows the relative concentration ratios ($C_t/C_0$) of TC versus the reaction time ($t$) in the systems with PDS presence. Within 2 h of the reaction time, the concentrations of TC sharply decreased. Then, they leveled off with reaction time. Although PDS was considered to be an excellent oxidative agent, it alone could only degrade about 18% of TC, which was consistent with the previous results that showed that PDS without proper activation was not efficient for the degradation of organic contaminants [46]. It is worth noting that, when SBC and PDS were added to the systems together, the removal efficiencies of TC in SBC-PDS systems were significantly increased, compared with that of a system with PDS alone. In particular, when the pyrolysis temperatures of SBCs raised from 300 °C to 700 °C, the removal rates of TC in the mixture systems were 78.2% of SBC300-PDS, 86.5% of SBC500-PDS, and 90.3% of SBC700-PDS, which were 4.3, 4.8, and 5.0 times that of the system with PDS alone. In our previous study [39], it was found that the total removal adsorption efficiencies of TC by SBCs were in the order of SBC300 (38.8%) < SBC500 (50.4%) < SBC700 (54.1%), which were 2.0, 1.7, and 1.7 times less than the removal rates of TC in the SBC300-, SBC500-, and SBC700-PDS system, respectively (Figure 2). SBCs could effectively activate PDS for rapid TC degradation. Furthermore, the higher the pyrolysis temperatures of biochars, the more obvious the activation and degradation performance.

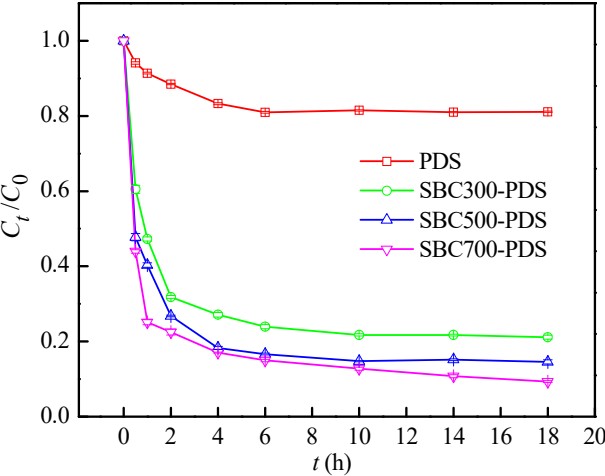

**Figure 2.** Relative concentration ratios ($C_t/C_0$) of TC versus reaction time ($t$) in the systems of PDS alone and SBC-PDS presence.

### 3.3. Mechanisms on PDS Activation by SBCs
#### 3.3.1. Free Radical Pathway

In order to clarify the oxidative radicals in SBC-PDS systems, chemical scavengers, MeOH and TBA, were chosen to identify the contribution of oxidative radicals such as $SO_4^{\bullet-}$ and $HO^{\bullet}$ to TC degradation. It has been recorded that MeOH could quench both $SO_4^{\bullet-}$ and $HO^{\bullet}$ with high reaction rates, $k = (1.6{\sim}7.7) \times 10^7\ M^{-1}\ S^{-1}$ and $k = (1.2{\sim}2.8) \times 10^9\ M^{-1}\ S^{-1}$ respectively, whereas TBA exhibited more sensitive quenching ability for $HO^{\bullet}$ ($k = (3.8{\sim}7.6) \times 10^8\ M^{-1}\ S^{-1}$) than for $SO_4^{\bullet-}$ ($k = (4{\sim}9.1) \times 10^5\ M^{-1}\ S^{-1}$) [47,48]. However, the quenching effect was not obviously observed in the present study. As shown

in Figure 3, the effects of MeOH and TBA addition on TC degradation patterns in SBC-PDS systems were insignificant, and the removal efficiencies were slightly different. When the molar ratio of MeOH to TC was 5000:1, the removal efficiencies of TC at 18 h of reaction time were 65.4%, 77.3%, and 79.2% in SBC300-, SBC500-, and SBC700-PDS systems, which decreased by 16.0%, 9.9%, and 10.5%, respectively, contrasted to 77.9%, 85.8%, and 88.5% with no MeOH addition. Correspondingly, the decreased removal efficiencies by TBA, at 5000:1 of the molar ratio of TBA to TC, were 12.1%, 7.7%, and 9.3%. The effects of TBA on removal efficiencies were less significant than those of MeOH. Moreover, the inhibitory effects of MeOH and TBA on TC removal weakened when their addition amounts decreased. The results show that MeOH and TBA could not terminate the oxidation of TC in SBC-PDS systems even at a high molar ratio of 5000: 1, suggesting that both $SO_4^{\bullet-}$ and $HO^{\bullet}$ partook in the TC degradation in SBC-PDS systems, but their contributions were somewhat lower. This indicates that the radical pathway was not the dominant one.

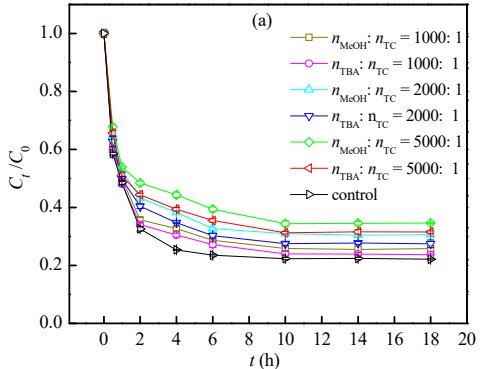
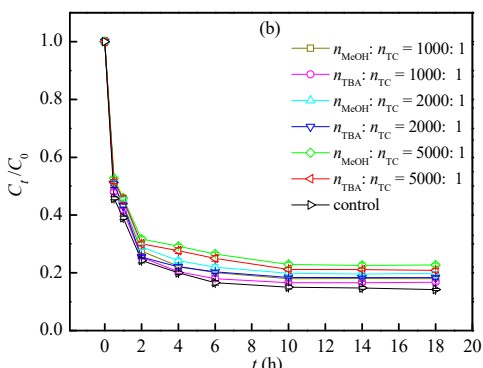

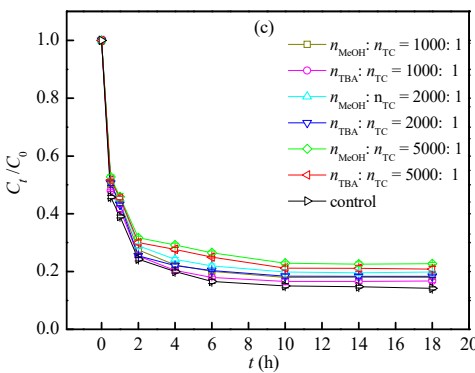

**Figure 3.** Effects of MeOH and TBA addition on relative concentration ratios ($C_t/C_0$) of TC versus reaction time (*t*) in SBC300 (**a**), SBC500– (**b**) and SBC700-PDS (**c**) systems.

### 3.3.2. Non-Radical Pathway

A few of the experimental results showed that the active radicals, $SO_4^{\bullet-}$ and $HO^{\bullet}$, did not always play the main roles in the organic pollutant degradation in the systems of activated persulfate by biochars but were usually dominated by non-free-radical species, such as singlet oxygen ($^1O_2$), and by electron transfer. Yin et al. prepared sludge biochar (SDBC) to inspire persulfate for the sulfamethoxazole degradation and found that singlet oxygen ($^1O_2$), rather than $HO^{\bullet}$ and $SO_4^{\bullet-}$, was the main reactive particulate in the SDBC/PDS system [25]. $^1O_2$ played a critical role in the degradation of bisphenol A (BPA) [27], and sulfathiazole (STZ) was degraded through a non-radical route through electron transfer [28]. Sun et al. used $CO_2$ to activate cellulose biochar and used it as a persulfate catalyst for the degradation of phenols, and the results showed that this new biochar/persulfate sys-

tem worked through a non-radical mechanism, including $^1O_2$ generation and electron transfer [48].

In this experiment, L-histidine was used as a singlet oxygen quencher ($k$ = ~3 × $10^7$ $M^{-1}$ $S^{-1}$) to explore whether singlet oxygen was the dominant factor for TC degradation in SBC-PDS systems [40,47]. As shown in Figure 4a–c, the effects of L-histidine addition on TC degradation patterns in SBC-PDS systems were insignificant, but the removal efficiencies of TC were quite different. When the concentration of L-histidine was 5 mmol/L, the removal rates of TC were 69.7%, 73.8%, and 71.4% in SBC300-, SBC500-, and SBC700-PDS systems, decreasing by 11.6%, 13.7%, and 21.3% compared with 78.9%, 84.9%, and 90.7% in SBC300-, SBC500-, and SBC700-PDS systems without the L-histidine addition. In the presence of 15 mmol/L L-histidine, the removal rates of TC decreased largely by 21.2%, 21.4%, and 32.2%, respectively, in SBC300-, SBC500-, and SBC700-PDS systems compared with those without the L-histidine addition. The results show that $^1O_2$ seemed to be the predominant oxidation particulate for the TC degradation, excluding the free radical $SO_4^{\bullet-}$ and $HO^\bullet$ produced in PDS activation and PDS direct oxidation, which indicates that the SBC can effectively inspire PDS activation to degrade TC through the $^1O_2$ oxidation route. As for the generation of $^1O_2$, there were two possible ways: (i) the recombination of $O_2^{\bullet-}$ and the reaction of $O_2^{\bullet-}$ and $HO^\bullet$ based on Equations (1) and (2) [49]; (ii) the generation of $^1O_2$ from the ketonic groups [50].

$$O_2^{\bullet-} + O_2^{\bullet-} + 2H^+ \rightarrow {}^1O_2 + H_2O_2 \tag{1}$$

$$O_2^{\bullet-} + HO^\bullet \rightarrow {}^1O_2 + OH^- \tag{2}$$

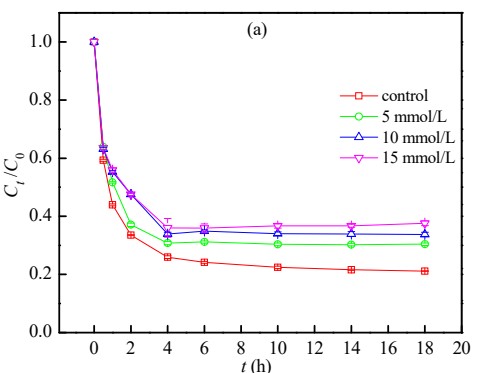
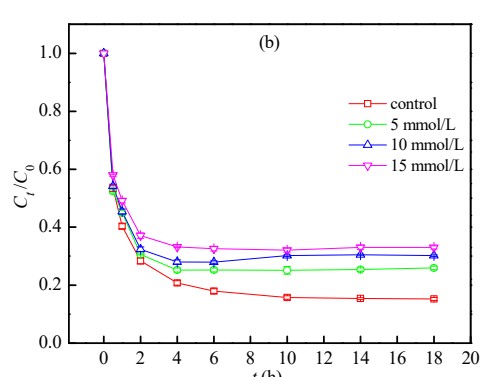

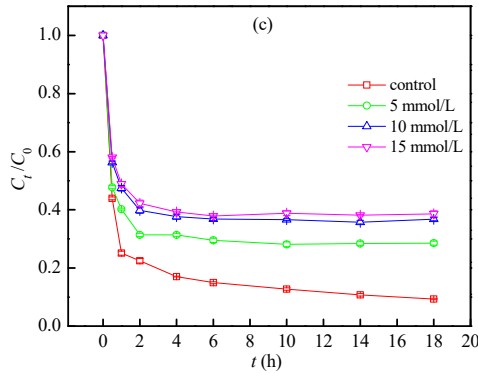

**Figure 4.** Effects of L-histidine addition on relative concentration ratios ($C_t/C_0$) of TC versus reaction time ($t$) in SBC300- (**a**), SBC500- (**b**) and SBC700-PDS (**c**) systems.

The electron transfer pathway from the target pollutants to the persulfate anions adjusted by carbon-based catalysts has been verified. Duan et al. found that carbon catalysts were supposed to improve PDS to form a surface complex which was capable of oxidizing the organics via a direct electron transfer [51]. To further investigate whether the electron transfer mechanism enhanced the oxidative degradation process in the systems, electrochemical tests such as linear-sweep voltammograms (LSV) and electrochemical impedance spectroscopy (EIS) were carried out. The changing of linear currents versus the potential of the two systems is plotted in Figure 5a–c. The results display that there was a certain current intensity in SBC-PDS+TC systems and systems with SBC alone, while the current intensity of SBC-PDS+TC systems was much greater than that of systems with SBC alone, which also indicates that the addition of oxidants and pollutants enhanced the current intensity SBC-PDS+TC. The current intensity of SBC700-PDS+TC system was higher than those of SBC500-PDS+TC and SBC300-PDS+TC systems. The results indicate that TC could be oxidized through non-radical electron-transfer mechanism. It was found that the graphitized structure of biochar could be used as an electron donor, because the activation was enhanced, and the proportion of oxygen-containing functional groups was lower at higher pyrolysis temperatures [52]. In addition, some studies have found that due to the graphitized carbon ($sp^2$ carbon), biochar would also be an electron transport mediator to promote the electron transfer [48].

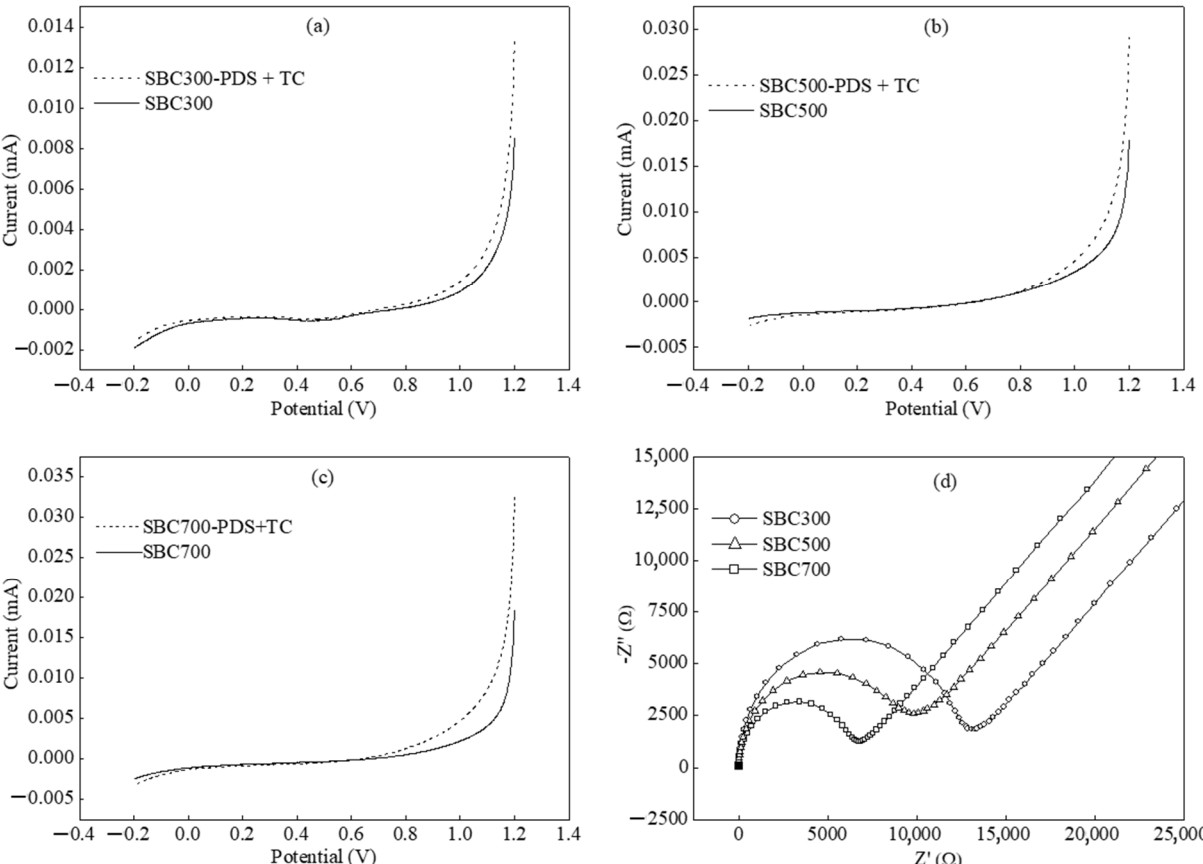

**Figure 5.** Linear-sweep voltammograms for SBC and SBC-PDS+TC systems ((**a**), SBC300; (**b**), SBC500; (**c**), SBC700) and electrochemical impedance spectroscopic analysis of SBCs loaded on the carbon paper electrodes (**d**).

Electrochemical impedance spectroscopic analysis (EIS) was used to study the charge transfer process in electrode materials. Figure 5d showed the Nyquist plots of SBC prepared at different pyrolysis temperatures in the three-electrode systems. The diameter of the semicircle indicated the resistance of the electrode material. The larger the diameter,

the greater the impedance; otherwise, the smaller the impedance [53]. Compared with SBC300 and SBC500, the impedance of SBC700 was the smallest, which indicates that higher pyrolysis could induce higher degrees of graphitization and conductivity, which was beneficial to the electron transfer.

Recently, the SS-derived biochars were used in the persulfate activation to degrade the refractory organic pollutants in wastewater. Because of the different components of sludge, the SS-derived biochars may exhibit different mechanisms, such as radical and nonradical pathways in persulfate. It was found that $SO_4^{\bullet-}$ and $HO^\bullet$ were responsible for the degradation of 4-chlorophenol [26], $^1O_2$ played a critical role in the degradation of bisphenol A and sulfamethoxazole [25,27], sulfathiazole was degraded by a non-radical pathway through electron transfer [28], and $SO_4^{\bullet-}$, $HO^\bullet$, and $^1O_2$ contributed to triclosan degradation [29]. It was found that $HO^\bullet$, $SO_4^{\bullet-}$, $^\bullet O_2^-$, $^1O_2$, and free electrons all partook in the bisphenol A removal process, and the non-radical pathway of the electron transfer predominated the reaction [54].

### 3.4. Optimum Conditions for Degradation of TC

The optimization of the catalyst dosage is very important for its application in wastewater treatment. Table 2 shows the effects of the activator (SBC) dosage on TC degradation. When the dosage started at 0.5 g/L and increased to 3.0 g/L, the removal rates of TC in SBC-PDS systems were increased successively. For example, the removal rates of TC in SBC700-PDS systems increased from 76.0% at 0.5 g/L of SBC700 to 95.9% at 3.0 g/L of SBC700. The further addition of SBCs could not achieve more ideal removal efficiency. It can be interpreted that although a high dosage of SBC might supply more active sites for PDS activation, the usage efficiency decreased when the number of active sites increased. Meanwhile, more free radicals were produced, and their self-quenching could cut down the reaction efficiency in the system [55]. From the point of view of electron transfer, when the number of electron donors (TC) and electron receptors (PDS) in the system became the limiting factor, the removal rate of TC in SBC-PDS systems would not change with an increase in the SBC dosage.

**Table 2.** Effects of biochar dosage, PDS dosage, initial solution pH value, and biochar reuse run on removal rate (% $\pm$ SD) of TC in SBC300-, SBC500- and SBC700-PDS systems ($t$ = 18 h).

| Condition | | SBC300-PDS | SBC500-PDS | SBC700-PDS |
|---|---|---|---|---|
| Biochar dosage (g/L) | 0.5 | 62.2 $\pm$ 0.97 | 72.8 $\pm$ 1.21 | 76.0 $\pm$ 0.32 |
| | 1.0 | 78.9 $\pm$ 0.54 | 85.7 $\pm$ 0.96 | 90.7 $\pm$ 0.23 |
| | 2.0 | 80.6 $\pm$ 0.23 | 87.6 $\pm$ 0.09 | 92.5 $\pm$ 0.41 |
| | 3.0 | 82.6 $\pm$ 0.31 | 89.8 $\pm$ 0.18 | 95.9 $\pm$ 0.64 |
| PDS dosage ($n_{PDS}$: $n_{TC}$) | 50:1 | 66.6 $\pm$ 0.39 | 74.2 $\pm$ 0.79 | 83.8 $\pm$ 0.23 |
| | 100:1 | 78.9 $\pm$ 0.54 | 85.7 $\pm$ 0.96 | 90. 7 $\pm$ 0.23 |
| | 200:1 | 80.4 $\pm$ 0.47 | 86.8 $\pm$ 0.55 | 91.7 $\pm$ 0.55 |
| | 300:1 | 79.3 $\pm$ 0.54 | 86.8 $\pm$ 0.55 | 91.8 $\pm$ 0.41 |
| Initial pH value | 3 | 82.8 $\pm$ 0.77 | 87.9 $\pm$ 0.70 | 94.1 $\pm$ 0.67 |
| | 5 | 79.7 $\pm$ 0.54 | 83.1 $\pm$ 0.55 | 88.2 $\pm$ 0.54 |
| | 7 | 78.9 $\pm$ 0.54 | 85.7 $\pm$ 0.96 | 90.7 $\pm$ 0.11 |
| | 9 | 78.0 $\pm$ 0.23 | 84.3 $\pm$ 0.67 | 87.6 $\pm$ 0.41 |
| | 11 | 59.2 $\pm$ 0.41 | 71.9 $\pm$ 0.81 | 83.8 $\pm$ 0.64 |
| Biochar reuse (run) | 1 | 79.5 $\pm$ 0.87 | 86.6 $\pm$ 0.79 | 89.4 $\pm$ 0.72 |
| | 2 | 74.2 $\pm$ 0.76 | 82.5 $\pm$ 0.69 | 84.7 $\pm$ 0.77 |
| | 3 | 66.4 $\pm$ 0.58 | 72.1 $\pm$ 0.32 | 76.9 $\pm$ 0.96 |
| | 4 | 52.8 $\pm$ 0.77 | 63.5 $\pm$ 0.55 | 71.5 $\pm$ 0.69 |

The PDS concentration affects the degradation of organic pollutants. Table 2 also shows the effects of the PDS dosage on the oxidative degradation of TC in the SBC-PDS systems. When the molar ratios of PDS to TC ranged from 50:1 to 100:1, the removal rates

of TC in the systems were markedly improved. However, with the further addition of PDS, the removal efficiency of TC tended to be leveled off. When the molar ratio of PDS to TC was 300: 1, the removal efficiencies of TC in the SBC-PDS systems were finally stabilized at 79.3% (SBC300), 86.8% (SBC500), and 91.8% (SBC700), respectively. It was demonstrated that, due to the already full use of SBC, the possibly excessive addition of PDS would not be activated to improve the removal efficiency.

Generally, the pH value of the solution plays a significant role in biochar's activating properties. The effects of the initial solution pH values on the removal of TC are shown in Table 2. The results show that the removal rates of TC were the highest in SBC300-, SBC500-, and SBC-700-PDS systems when the solution pH value was 3, which were 83.0%, 87.8%, and 94.4% respectively. When the pH values increased, the removal rates of TC decreased gradually. TC has three fraction species at different pH values: pH < 3.3, the protonated form ($TCH_3^+$); 3.3 < pH < 7.7, the neutral form ($TCH_2^0$); and pH > 7.7, the monoanionic form ($TC^-$) [56]. Therefore, the above results might be attributed to two reasons: (i) the changed electrostatic force between SBC and PDS. The pH value of the solution could change the surface charge of SBC. Specifically, when the pH values were lower than the $pH_{PZC}$ values (7.98 of SBC300, 9.25 of SBC500, and 9.87 of SBC700), SBCs showed positively charged surfaces, where the negative ions of PDS ($S_2O_8^{2-}$) could get to the surfaces of SBCs easily by electrostatic attraction, which could enhance the pollutant oxidation via a direct electron transfer through the carbon matrixes [52]; (ii) considering the electron transfer effect, the electrostatic attraction between the biochar, which was used as the carrier of electronic transport and the electronic receptor (PDS), could effectively promote the electron transfer process under the low pH value, and then enhance the removal effect of TC. At large, the removal of TC in SBC-PDS systems could be obtained in a wide range of pH values.

Reusability is an important aspect regarding the evaluation of the catalytic performance of biochar. It is found in Table 2 that with the increased reuse runs, the removal efficiencies of TC in the SBC-PDS systems were gradually weakened. After four times of reuse, the removal efficiency of TC in SBC300-, SBC500-, and SBC700-PDS systems decreased by 26.7%, 23.1% and, 17.9%, respectively. However, SBC700 exhibited better reusability with higher TC removal efficiencies after four runs.

## 4. Conclusions

Compared to SBC300 and SBC500, SBC700 demonstrated a higher removal efficiencies of TC in SBC-PDS systems, which could be due to SBC700's larger specific surface area and more aromatic properties, resulting from higher pyrolysis temperatures. The free radicals ($SO_4^{\bullet-}$ and $HO^\bullet$), singlet oxygen ($^1O_2$), and electron transfer contributed to the degradation of TC in the SBC-PDS systems. However, it seemed that a non-radical pathway (i.e., singlet oxygen ($^1O_2$) and electron transfer) was the main mechanism for TC degradation. The removal efficiencies of TC in the SBC-PDS systems could be up to 96.0% (SBC700-PDS) under the optimum conditions. SBC700 could be superior to the municipal sewage sludge-derived biochar and an effective catalyst to activate PDS for the degradation removal of TC in wastewater.

**Author Contributions:** Conceptualization, B.Z.; methodology, B.Z.; software, J.Z.; validation, B.Z. and J.Z.; formal analysis, B.Z. and J.Z.; investigation, J.Z.; resources, B.Z.; data curation, J.Z.; writing—original draft preparation, J.Z.; writing—review and editing, B.Z.; visualization, B.Z.; supervision, B.Z.; project administration, B.Z.; funding acquisition, B.Z. All authors have read and agreed to the published version of the manuscript.

**Funding:** This research was funded by the National Natural Science Foundation of China, grant numbers 51766008, 21467013, and 21167007.

**Institutional Review Board Statement:** Not applicable.

**Informed Consent Statement:** Not applicable.

**Data Availability Statement:** Not applicable.

**Acknowledgments:** The authors are sincerely grateful to the testing center for providing technical support.

**Conflicts of Interest:** The authors declare no conflict of interest.

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
