# Peer review of "Tetracycline Degradation by Peroxydisulfate Activated by Waste Pulp/Paper Mill Sludge Biochars Derived at Different Pyrolysis Temperature"

_water, doi:10.3390/w14101583_

Round 1
Reviewer 1 Report
This article is interesting and well crafted. Interesting to use PPS instead of SS.
The introduction describes that PPS content of cellulose, hemicellulose, lignin, etc. can make the biochar beneficial for the purpose.
I would like to read a comment, in the discussion or conclusion, if it turned out to be beneficial compared to SS.
Author Response
Response to the Review report 1
MS No. water-1666993
Title: Tetracycline Degradation by Peroxydisulfate Activated by Waste Pulp/Paper Mill Sludge Biochars Derived at Different Pyrolysis Temperature
This article is interesting and well crafted. Interesting to use PPS instead of SS.
Response: Thanks to the comment.
The introduction describes that PPS content of cellulose, hemicellulose, lignin, etc. can make the biochar beneficial for the purpose.
I would like to read a comment, in the discussion or conclusion, if it turned out to be beneficial compared to SS.
Response: Biochar may originate from a highly diverse range of feedstocks including, but not limited to, forestry products, agricultural residues, animal wastes or municipal waste products. In most cases, the original biomass structure strongly influences the final biochar structure, its eventual physical characteristics, and its eventual reaction in different systems. The most commonly observed feature in processed biochar is a pore or tube-like structure, together with larger specific surface area, that mimics the cellular structure of wood or plant-based feedstocks with high content of hemicellulose, cellulose and lignin. PPS mainly originates from wood or plant-based biomass. Thus, the PPS-derived biochars could be superior to the SS-derived biochars in view of the structure and specific surface area. For example, the specific surface areas of the PPS-derived biochars in this study, SBC300, SBC500 and SBC700, are 7.12, 32.82 and 37.80 m2/g, which are larger than those of the SS-derived biochars, 5.11 m2/g for BC300, 15.23 m2/g for BC500 and 13.57 m2/g for BC700, obtained in our previous study [Environmental Technology & Innovation 26 (2022) 102288]. Therefore, Compared with SS, the waste pulp/paper mill sludge (PPS) is characterized with high content of hemicellulose, cellulose and lignin [30], which could be beneficial to the derived biochars to transfer electrons to metals to produce free radicals [31,32], to improve the generation of reactive oxygen species (ROS) (SO4•−, HO•, O2•−, and 1O2) and would have an important effect on the catalytic degradation of organic contaminants [33].
Supplement has been done in Conclusion.

Reviewer 2 Report
This paper has reported the application of biochar derived from waste pulp/paper mill sludge in wastewater treatment, i.e. tetraccycline degradation. The authors have put a lot of effort generating new summarization. The manuscript is of certain novelty and the work carried out is creditable. The manuscript needs further revision before it can be considered to accept in Water journal.
The comments are given as follows:
1) XRD pattern can be enlarged to display the peaks clearer.
2) BET data is available in the manuscript but there is no explanation in relation to the data, especially for temperature effect on surface area of biochar.
3) Biochar characteristic and treatment efficiency are dependent on synthesis procedure. Some recommendation on synthesis procedure should be provided. Besides, which parameter of analysis is the most determining factor for biochar application should be concerned.
4) Lack of biochar preparation procedure. Please describe in details.
5) How is about the effect of major elements (C, O, N and S) on PDS activation?
6) The results of this study should be compared with the previous studies.
7) Grammatical errors should be avoided. Quality of figures also needs to be improved.
Author Response
Response to the Review report 2
MS No. water-1666993
Title: Tetracycline Degradation by Peroxydisulfate Activated by Waste Pulp/Paper Mill Sludge Biochars Derived at Different Pyrolysis Temperature
This paper has reported the application of biochar derived from waste pulp/paper mill sludge in wastewater treatment, i.e. tetraccycline degradation. The authors have put a lot of effort generating new summarization. The manuscript is of certain novelty and the work carried out is creditable. The manuscript needs further revision before it can be considered to accept in Water journal.
Response: Thanks to the reviewer.
The comments are given as follows:
- XRD pattern can be enlarged to display the peaks clearer.
Response: The XRD pattern has been enlarged.
2) BET data is available in the manuscript but there is no explanation in relation to the data, especially for temperature effect on surface area of biochar.
Response: Thanks to the reviewer. The description and explanation have been added, as “The specific areas of biochars were 7.12, 32.82 and 37.80 m2/g, respectively. ……The escape of generated gas and volatile components caused the creation of pore structure in the solid. Upon raise in pyrolytic temperature, the tar was gradually decomposed, the pore formation rate was accelerated, and the specific surface area increased [30].” in the revised manuscript.
3) Biochar characteristic and treatment efficiency are dependent on synthesis procedure. Some recommendation on synthesis procedure should be provided. Besides, which parameter of analysis is the most determining factor for biochar application should be concerned.
Response: For the procedure of biochar preparation, we did not design and conduct temperature programming. The sludge was naturally dried, ground, and sifted through a 60 mesh sieve. Then the sludge powder was put into a muffle furnace and pyrolyzed at 300, 500 and 700℃ for 6 h by oxygen-limiting and temperature-controlling methods. After cooling to room temperature, the biochars were taken out and stored, and recorded as SBC300, SBC500 and SBC700, respectively. We are sorry. No correction was done.
4) Lack of biochar preparation procedure. Please describe in details.
Response: As mentioned above, the sludge was naturally dried, ground, and sifted through a 60 mesh sieve. Then the sludge powder was put into a muffle furnace and pyrolyzed at 300, 500 and 700℃ for 6 h by oxygen-limiting and temperature-controlling methods. After cooling to room temperature, the biochars were taken out and stored, and recorded as SBC300, SBC500 and SBC700, respectively. We are sorry. No correction was done.
5) How is about the effect of major elements (C, O, N and S) on PDS activation?
Response: Thanks to the reviewer. Surely, the previously studies have reported that the surface properties of biochar may be responsible for persulfate activation. The –OH and C=O groups have been identified as the active sites to general reactive free radical such as hydroxyl radical and sulfate radical. Also, it has been reported that N (pyridinic N and graphitic N) could promote the electron transfer at the interface of carbon-based catalysts.
The corresponding description has been added in the revised manuscript. “It has been shown that some important functional groups consisting of carbon, hydrogen and oxygen atoms (for example, -OH and C=O) are the active sites to general reactive free radical such as hydroxyl radical and sulfate radical. It has been indicated that nitrogen atom (pyridinic N and graphitic N) could enhance the electron transfer at the interface of carbon-based catalysts [42].”
6) The results of this study should be compared with the previous studies.
Response: Some comparison was done as “Recenly, the SS-derived biochars were used in the persulfate activation to degrade the refractory organic pollutants in waste water. Because of the different components of sludge, the SS-derived biochars may exhibit different mechanisms such as radical and nonradical pathways in persullfate. It was found that SO4•− and HO• were responsible for the degradation of 4-chlorophenol [26], 1O2 played a critical role in the degradation of bisphenol A and sulfamethoxazole [25,27], sulfathiazole was degraded by a non-radical pathway through electron transfer [28], and SO4•−, HO• and 1O2 contributed to triclosan degradation [29]. It was found that HO•, SO4•−, •O2 - , 1O2, and free electron all partook in bisphenol A removal process and the non-radical pathway of electron transfer predominated the reaction [54].”
7) Grammatical errors should be avoided. Quality of figures also needs to be improved.
Response: Correction has been done.

Reviewer 3 Report
The article deals with an important topic of removal of antibiotics from wastewater by an example of tetracycline. The article itself is quite meticulous, and repeating of the experiments is very commendable as well. The results of article may be used in medicine and biotechnology, however, the current version of the article requires significant changes.
- Abstract requires a rewriting because currently it is a mess of acronyms and abbreviations and it is difficult to comprehend the main principal findings of the article.
- Line 55-58, 62 and may be others. No reason to use abbreviations for compounds that are used in the article only once.
- Line 61. It’s not clear what transfer of electrons to metal has to do with the generation of reactive oxygen species.
- Line 68. What’s an EDA?
- Line 84. Every chemists know what HCl and NaOH are.
- Line 128-129. The additives are defined by the ratio (2000:1 etc.) and by the molar concentration. What is the reason for the inconsistency? It would be easier to understand if the units were the same in both cases.
- Line 150. Is elemental composition given in molar % or mass %? Also, how H/C is calculated? Because for SBC300 19.98/1.95 doesn’t give 0.66 (and the same is for other samples). Moreover, it looks like it follows the pattern of 1/1.5, 1/3, 1/4. O/C and (N+O)/C ratios are calculated correctly though.
- Line 185, 206, 239. Different removal efficiencies of TC. Is it because the authors made control experiments every time before the addition of new compound? If yes then you should specify that. If no then why the removal rates are different?
- Line 259. The authors made experiments with SBC alone, and these carbon-based materials can absorb TC. What is the sorption rate of TC to SBC? I think the authors should separate two effects – removal of TC from water and degradation of TC. Were there any experiments when TC degraded without addition of SBC or PDS? If the authors didn’t perform such experiments, at least, comparison with the literature data should be made.
- Line 261. The authors say that “the addition of oxidants and pollutants enhanced the current”. Going to lines 109-111, I can calculate the following: 50 mg in 50 ml of solution equals to 1 g/l of SBC (probably, dispersion). 20 mg/L TC equals to 4.5*10E-5 moles of TC in 1 L of solution. Sodium PDS content is 100 times higher than TC, therefore it is 4.5*10E-4 moles in 1 L of solution. Did the author consider the increase of conductivity of the solution? Was it measured/detectable?
- Line 287. In continuation of question 9, how did the authors become certain that the removal of TC is because of activation of PDS sites and not because of sorption of TC in carbon-based material?

Author Response
Response to the Review report 3
MS No. water-1666993
Title: Tetracycline Degradation by Peroxydisulfate Activated by Waste Pulp/Paper Mill Sludge Biochars Derived at Different Pyrolysis Temperature
The article deals with an important topic of removal of antibiotics from wastewater by an example of tetracycline. The article itself is quite meticulous, and repeating of the experiments is very commendable as well. The results of article may be used in medicine and biotechnology, however, the current version of the article requires significant changes.
Response: Thanks to the reviewer.
- Abstract requires a rewriting because currently it is a mess of acronyms and abbreviations and it is difficult to comprehend the main principal findings of the article.
Response: Thanks to the reviewer. We avoid the use of acronyms and abbreviations as can as possible, such as PS, TC, PPS and PDS. However, those for biochars, i.e. SBC300, SBC500 and SBC700, are still used because “the pulp/paper mill sludge biochars” is too long to represent the biochars.
- Line 55-58, 62 and may be others. No reason to use abbreviations for compounds that are used in the article only once.
Response: Thanks to the reviewer. Corrections have been made.
- Line 61. It’s not clear what transfer of electrons to metal has to do with the generation of reactive oxygen species.
Response: We are sorry for the ambiguous description. Correction has been made.
- Line 68. What’s an EDA?
Response: We are sorry for the direct use of abbreviation EDA. It refers as “electron donor-acceptor”. Correction has been made.
- Line 84. Every chemists know what HCl and NaOH are.
Response: The common reagents HCl and NaOH were deleted.
- Line 128-129. The additives are defined by the ratio (2000:1 etc.) and by the molar concentration. What is the reason for the inconsistency? It would be easier to understand if the units were the same in both cases.
Response: For the free radical quenching experiments, two kinds of scavengers were used, methanol (MeOH) and tert-butyl alcohol (TBA). If the molar concentrations were used, there would be three concentrations for MeOH and three concentrations for TBA. The corresponding concentrations of MeOH and TBA at the same ratios would be different, which is not easy to compare their quenching effect. For the singlet oxygen quenching experiments, only one scavenger, L-histidine, was used. So, the molar concentrations were used. No correction was made.
- Line 150. Is elemental composition given in molar % or mass %? Also, how H/C is calculated? Because for SBC300 19.98/1.95 doesn’t give 0.66 (and the same is for other samples). Moreover, it looks like it follows the pattern of 1/1.5, 1/3, 1/4. O/C and (N+O)/C ratios are calculated correctly though.
Response: The elemental composition is given in mass %. The atomic ratios are given in molar ratios. We are so sorry for our carelessness in the Excell calculation. The corrections have been made.
- Line 185, 206, 239. Different removal efficiencies of TC. Is it because the authors made control experiments every time before the addition of new compound? If yes then you should specify that. If no then why the removal rates are different?
Response: The authors did not make control experiments. The three groups of removal data are (78.2%, 86.5%, 90.3%), (77.9%, 85.8%, 88.5%) and (78.9%, 84.9%, 90.7%) respectively, which were obtained at the same experimental conditions. The authors think the minor differences among them are normal experimental errors. No correction was made.
- Line 259. The authors made experiments with SBC alone, and these carbon-based materials can absorb TC. What is the sorption rate of TC to SBC? I think the authors should separate two effects – removal of TC from water and degradation of TC. Were there any experiments when TC degraded without addition of SBC or PDS? If the authors didn’t perform such experiments, at least, comparison with the literature data should be made.
Response: The authors have done the adsorption experiments. The adsorption results have been published in the Ref. [39], where the total adsorption removal efficiencies of TC by SBCs were in the order of SBC300 (38.8%) < SBC500 (50.4%) < SBC700 (54.1%). In Figure 2, the removal rates of TC in the SBC-PDS systems were 78.2% of SBC300-PDS, 86.5% of SBC500-PDS and 90.3% of SBC700-PDS, which were 2.0, 1.7 and 1.7 times that in SBC300, SBC500 and SBC700 alone system. The comparison on this has been supplemented in the revised paper as “In our previous study [39], it was found that the total removal adsorption efficiencies of TC by SBCs were in the order of SBC300 (38.8%) < SBC500 (50.4%) < SBC700 (54.1%), which were 2.0, 1.7 and 1.7 times less than the removal rates of TC in the SBC300-, SBC500- and SBC700-PDS system, respectively (Figure 2)”.
- Line 261. The authors say that “the addition of oxidants and pollutants enhanced the current”. Going to lines 109-111, I can calculate the following: 50 mg in 50 ml of solution equals to 1 g/l of SBC (probably, dispersion). 20 mg/L TC equals to 4.5*10E-5 moles of TC in 1 L of solution. Sodium PDS content is 100 times higher than TC, therefore it is 4.5*10E-4 moles in 1 L of solution. Did the author consider the increase of conductivity of the solution? Was it measured/detectable?
Response: Thanks to the reviewer. Surely, the addition of TC and PDS would increase the conductivity of the solution. If there was no oxidation reaction, the increase extent of current by the addition of TC and PDS would be the same in the SBC-PDS systems. However, because the electron transfer extents are different in the SBC-PDS + TC systems, the increase extents of current in the SBC-PDS + TC systems are different (Figure 5). No correction was made.
- Line 287. In continuation of question 9, how did the authors become certain that the removal of TC is because of activation of PDS sites and not because of sorption of TC in carbon-based material?
Response: Thanks to the reviewer. As responded to the question 9, if there is no PDS, the TC could be partially removed by the adsorption of SBCs. However, in the presence of PDS, the activation of PDS by biochars and the oxidation reaction occurred, which could be proved by the experiments for mechanism exploration. No correction was made.

Round 2
Reviewer 2 Report
The authors have addressed all comments. This manuscript can be accepted.
Author Response
Response to the Review report 2 (Round 2)
MS No. water-1666993
Title: Tetracycline Degradation by Peroxydisulfate Activated by Waste Pulp/Paper Mill Sludge Biochars Derived at Different Pyrolysis Temperature
The authors have addressed all comments. This manuscript can be accepted.
Response: Thanks to the reviewer.

Reviewer 3 Report
Current version of the article is improved. The authors answered almost all of the issues and questions, but several new arose. Thankfully, the new issues are relatively minor, and in my opinion, the article can be published in “Water” after fixing them.
I also want to point out that I have marked the question “Are the results clearly presented?” as “must be improved”. It is not because the results are bad (they are good) but because it is very hard to read the data from the figures.
Line 4. Missing the ending of the title. Also, the authors may think about shortening the title.
Lines 9-24. The abstract now looks better. In my opinion, it is quite long, but that’s up to authors if they want to shorten it.
Lines 128-129. I still don’t understand what is the reason to use molar concentration for L-histidine. It is possible to use ratios instead molarities for singlet oxygen quenching experiment, so the readers could compare different quenching approaches. But if the authors feel that it should be molar concentrations then I won’t argue.
Lines 137, 159. The authors must specify that the elemental composition is given in mass, not molar percent.
Line 197. Fig. 1 doesn’t have x-axis (angles, wavelength).
Lines 211, 248, 303. According to the authors comments, each series (or group) of the experiments were performed firstly without the addition of chemical scavengers. It is very commendable that the authors didn’t use the same value obtained in the initial experiment, but instead investigated the complete series every time. It also means that the repeatability of the experiments is high (the resulting efficiency changes for no more than +/- 1%). Also, Table 2 with 1 g/L biochar, 100:1 PDF, pH 7 shows the same efficiencies. I think the authors should indicate that in the article and also mention that the effect of scavengers is much higher.
Fig. 2, Fig. 3 and Fig. 4. Please, make the figures colored, because “Water” allows colored figures in text. It is especially difficult to see the data in Fig. 3.
Also, I suggest to authors to try and use logarithmic scale for Ct/C0, because in my opinion, the lines would be straight or close to straight.
Line 420. Please, rephrase “overmuchadded-PDS”
Line 433. What is “fication”?
Author Response
Response to the Review report 3 (Round 2)
MS No. water-1666993
Title: Tetracycline Degradation by Peroxydisulfate Activated by Waste Pulp/Paper Mill Sludge Biochars Derived at Different Pyrolysis Temperature
Current version of the article is improved. The authors answered almost all of the issues and questions, but several new arose. Thankfully, the new issues are relatively minor, and in my opinion, the article can be published in “Water” after fixing them.
Response: Thanks to the reviewer.
I also want to point out that I have marked the question “Are the results clearly presented?” as “must be improved”. It is not because the results are bad (they are good) but because it is very hard to read the data from the figures.
Response: Thanks to the reviewer. The figures have been replotted in color.
Line 4. Missing the ending of the title. Also, the authors may think about shortening the title.
Response: The missing ending of the title has been supplemented. Because the use of abbreviation for “Waste Pulp/Paper Mill Sludge Biochars” is not proper in the title, the length of title was not changed.
Lines 9-24. The abstract now looks better. In my opinion, it is quite long, but that’s up to authors if they want to shorten it.
Response: Thanks to the reviewer. No correction was made.
Lines 128-129. I still don’t understand what is the reason to use molar concentration for L-histidine. It is possible to use ratios instead molarities for singlet oxygen quenching experiment, so the readers could compare different quenching approaches. But if the authors feel that it should be molar concentrations then I won’t argue.
Response: Thanks to the reviewer. No correction was made.
Lines 137, 159. The authors must specify that the elemental composition is given in mass, not molar percent.
Response: Thanks to the reviewer. “wt%” has been indicated in Table 1 for the elemental composition.
Line 197. Fig. 1 doesn’t have x-axis (angles, wavelength).
Response: Thanks to the reviewer. The x-axis is presented in the original uploaded Word file. The Submission System automatically changed the file into PDF version. Thus, the version of PDF file is different from the Word one and the x-axis was sheltered. No correction was made.
Lines 211, 248, 303. According to the authors comments, each series (or group) of the experiments were performed firstly without the addition of chemical scavengers. It is very commendable that the authors didn’t use the same value obtained in the initial experiment, but instead investigated the complete series every time. It also means that the repeatability of the experiments is high (the resulting efficiency changes for no more than +/- 1%). Also, Table 2 with 1 g/L biochar, 100:1 PDF, pH 7 shows the same efficiencies. I think the authors should indicate that in the article and also mention that the effect of scavengers is much higher.
Response: Thanks to the reviewer. The ratio of “0:1” and the concentration of “0” mmol/L were added in the description for quenching experiments to present the series without the addition of chemical scavengers, as “For the free radical quenching experiments, the molar ratios of scavenger, methanol (MeOH) or tert-butyl alcohol (TBA), to TC were set as 0:1, 1000:1, 2000:1 and 5000:1 respectively. For the singlet oxygen quenching experiments, L-histidine was added with the concentrations of 0, 5, 10 and 15 mmol/L in the systems.” The much high effect of scavengers was indicated, as “the removal rates of TC decreased largely by 21.2%, 21.4% and 32.2% respectively in SBC300-, SBC500- and SBC700-PDS systems compared with those without L-histidine addition”.
Fig. 2, Fig. 3 and Fig. 4. Please, make the figures colored, because “Water” allows colored figures in text. It is especially difficult to see the data in Fig. 3.
Response: The figures were replotted in color.
Also, I suggest to authors to try and use logarithmic scale for Ct/C0, because in my opinion, the lines would be straight or close to straight.
Response: Thanks to the reviewer. Surely, if the logarithmic concentration versus reaction time was plotted, the lines would be straight or close to straight and the degradation kinetics could be viewed. However, this paper focused on the efficiencies, mechanisms and condition optimization of tetracycline degradation by PDS activated by biochars derived at different pyrolysis temperature. The degradation kinetics will be investigated in the future. We are sorry for no correction.
Line 420. Please, rephrase “overmuchadded-PDS”
Response: Correction has been made. Thanks!
Line 433. What is “fication”?
Response: We are sorry for the inaccurate word. It was deleted.
